# Prevalence of Cow’s Milk Allergy in Infants from an Urban, Low-Income Population in Chile: A Cohort Study

**DOI:** 10.3390/nu17111859

**Published:** 2025-05-29

**Authors:** Sylvia Cruchet, María Eugenia Arancibia, Andrés Maturana, Pamela Marchant, Lorena Rodríguez, Yalda Lucero

**Affiliations:** 1Instituto de Nutrición y Tecnología de los Alimentos, Universidad de Chile, Santiago 7830490, Chile; scruchet@inta.uchile.cl; 2Clínica Alemana de Santiago de Chile, Facultad de Medicina, Clínica Alemana-Universidad del Desarrollo, Santiago 7650568, Chile; marancibia@alemana.cl (M.E.A.); amaturana@alemana.cl (A.M.); lolyrodriguez19@gmail.com (L.R.); 3Hospital Padre Hurtado, Santiago 8880465, Chile; pamela.marchant.a@gmail.com; 4Department of Pediatrics and Pediatric Surgery, Hospital Roberto del Río, Facultad de Medicina, Universidad de Chile, Santiago 8380453, Chile

**Keywords:** food allergy, cow’s milk protein allergy, non-IgE mediated food allergy, infants

## Abstract

**Background.** Cow’s milk allergy (CMA) is one of the most common food allergies in infancy, with prevalence estimates of 0.5–7.5% in high-income countries. Data from low- and middle-income regions remain limited, and the predominant immune mechanism (IgE or non-IgE mediated) may vary across populations. **Objective.** We aimed to determine the prevalence and clinical characteristics of CMA in infants from an urban, low-income Chilean population. **Methods.** A prospective cohort study was conducted at Padre Hurtado Hospital in Santiago, Chile. Healthy term newborns were recruited and followed for up to 12 months. Sociodemographic, perinatal data and parental atopy were recorded. Parents were contacted monthly to screen for CMA symptoms. Infants with ≥two symptoms underwent clinical evaluation, a 4-week cow’s milk protein exclusion diet, and an open oral food challenge (OFC). Diagnosis followed international consensus guidelines. **Results.** Of 552 enrolled infants (48% male), 27 were diagnosed with CMA, yielding a prevalence of 4.9% (95% CI 3.1–7.0%). All cases exhibited non-IgE-mediated symptoms, including vomiting, dermatitis, colic, and perianal erythema. CMA was diagnosed before 6 months of age in 74% of cases. At 12 months, 40% had developed oral tolerance. Sociodemographic and perinatal characteristics were similar between groups, but some self-reported parental atopic traits were more frequent in CMA cases. **Conclusions.** CMA prevalence in this Chilean cohort was comparable to that reported in high-income countries, with a predominance of non-IgE-mediated forms. These findings support the need for standardized diagnostic protocols, including OFC, in diverse populations. Future studies should explore long-term outcomes and risk factors in non-IgE-mediated CMA.

## 1. Introduction

Food allergies (FAs) are a group of immune-mediated adverse reactions triggered by exposure to specific food proteins [1]. These conditions affect approximately 0.5–10% of children worldwide and have become a growing public health concern due to their rising prevalence, variable clinical presentations, and impact on quality of life and healthcare systems [2].

Cow’s milk proteins (CMPs) are the most reported food allergens [1,3]. Cow’s milk allergy (CMA) is a frequent condition, especially affecting infants, with an estimated prevalence ranging from 0.5% to 7.5% in high-income countries [1,4,5,6,7]. Clinical manifestations of CMA are heterogeneous, including cutaneous, respiratory, and especially gastrointestinal symptoms, such as vomiting, diarrhea, constipation, abdominal pain, and distension, which may overlap or be confounded with other infant disorders, including those resulting from gut–brain interaction [1,8,9]. In breastfed infants, symptoms of CMA may occur due to the transfer of CMP through breast milk. In such cases, the maternal dietary elimination of bovine dairy products is considered as a first-line intervention to confirm diagnosis and improve symptoms. This strategy is supported by international guidelines and highlights the importance of considering maternal diet when evaluating allergic symptoms in breastfed infants.

CMA is classified into IgE-mediated and non-IgE-mediated types, with mixed forms also reported [3]. Most European and North American studies indicate that IgE-mediated CMA predominates in those latitudes [4,5,6,7,10,11,12]. Blood or skin prick tests are typically used to identify specific IgE antibodies against food allergens for diagnosing IgE-mediated FAs [3]. However, evidence from Israel, Australia, Spain, and Japan suggests that non-IgE-mediated forms may be more prevalent in other regions [13,14,15]. Diagnosing non-IgE-mediated CMA poses substantial challenges due to the absence of validated biomarkers and reliance on clinical judgment based on symptom resolution after a strict 2- to 4-week CMP exclusion diet, followed by an oral food challenge (OFC) [1]. This approach is crucial to avoid both underdiagnosis and overdiagnosis, which can lead to unnecessary dietary restrictions or missed treatment opportunities.

Despite the worldwide growing interest in CMA, there is a critical lack of population-based studies using standardized diagnostic criteria, including OFCs, in low- and middle-income countries, limiting our understanding of its true burden and immunological profile in these settings. This is especially relevant in Latin America, where healthcare access, dietary patterns, and microbial exposures may differ significantly from other regions, potentially influencing both the prevalence and immunopathological profile of CMA.

This study aimed to establish the prevalence of CMA, confirmed according to international consensus guidelines, in a cohort of infants living in an urban, low-income area in the Metropolitan Region of Chile, a middle-income country in the Southern Cone of the Americas. In addition, we sought to describe the predominant clinical presentation pattern (whether suggestive of IgE- or non-IgE-mediated allergy) based on symptomatology, as well as the main clinical features at diagnosis and at one year of age.

## 2. Materials and Methods

### 2.1. Study Design and Setting

This cohort study included healthy, term newborns from the nursery of Padre Hurtado Hospital, a public healthcare facility in Santiago, Chile, that primarily serves a low-income urban population. The infants were followed up to one year of life.

### 2.2. Exclusion Criteria

Infants were excluded if they were born with a very low birth weight or had any serious condition during the first month of life, including sepsis, congenital heart disease, major malformations, genetic disorders, metabolic diseases, immunodeficiency, or any condition requiring surgical intervention. Infants were also excluded if their parents could not be contacted, had no available telephone number, or had plans to move outside the healthcare coverage region during the one-year follow-up period.

### 2.3. Patient Recruitment and Follow-Up

Consecutive cases were recruited after receiving informed consent from their parents until the sample size was reached. A coordinator nurse contacted the parents of recruited patients by phone every 4 weeks. In the first interview, the sociodemographic data of the patient and parents, perinatal history, and self-reported history of atopy from the parents were registered. The parents were asked monthly for general, gastrointestinal, dermatological, and respiratory symptoms suggestive of CMA according to a predefined checklist (Appendix A). If the patient had at least 2 suggestive manifestations, an experienced pediatric gastroenterologist evaluation was scheduled. In this evaluation, the physician performed a standardized anamnesis and physical examination.

### 2.4. Diagnostic Protocol for CMA

In cases considered suggestive of CMA, a 4-week CMP elimination diet was indicated. In breastfed infants, a maternal CMP-free diet was prescribed, in line with international recommendations. In formula-fed infants with mild to moderate symptoms, an extensively hydrolyzed formula was provided. In severe formula-fed cases or those persisting with symptoms after 4 weeks of an exclusion diet, a trial with an aminoacidic formula was prescribed.

After the 4-week exclusion diet, the pediatric gastroenterologist re-evaluated the patient. If symptoms had improved, an open-label OFC was conducted by reintroducing cow’s milk protein into the infant’s or mother’s diet, depending on the feeding modality. If symptoms reappeared during the subsequent 2 weeks, CMA was confirmed, and the patient remained on a CMP-free diet until one year of age. Otherwise, the diagnosis was ruled out. All infants were followed monthly by telephone until 12 months of age, using the same structured symptom checklist applied during the initial screening phase. Infants with confirmed CMA were re-evaluated in person by a pediatric gastroenterologist at one year of age, at which point an open OFC with CMP was indicated. A follow-up assessment was conducted one month later to determine the development of oral tolerance.

### 2.5. Sample Size Calculation and Statistical Analysis

Considering an estimated amount of 5500 potentially eligible healthy, term newborns, assuming a mean prevalence of 3% according to the literature review [16,17,18,19], with a 95% CI, a precision of 1.5%, and an expected loss of follow-up of 17.5%, a sample size of 550 individuals was calculated as representative of our population. Consecutive recruitment was conducted until the sample size was reached.

Codified data were loaded into a secure database. Data analysis was performed using SPSS software (version 22). Categorical variables are expressed as frequencies and percentages, and continuous variables are expressed in terms of central tendency and dispersion parameters. In order to compare categorical variables, the Chi-square test or the exact Fisher’s test was used, depending on the number of observations. To assess the normality of continuous variables, the Shapiro–Wilk test was used, and comparisons were performed using either the Student’s *t*-test or the Mann–Whitney U test, as appropriate. All statistical tests were two-tailed, and an alpha level of 0.05 was considered statistically significant.

## 3. Results

During enrolment, 1129 newborns were screened, of which 989 fulfilled the eligibility criteria. Finally, a total of 552 patients were successfully enrolled and followed up. Figure 1 shows the details of recruitment, the diagnostic process, and confirmed cases. Appendix A shows a summary of the loss of follow-up month by month. We had a 9.8% loss of follow-up during the first 6 months of life, a period when most of the CMA cases were diagnosed.

A total of 552 newborns entered the monthly telephone follow-up phase. Appendix A presents detailed information on the number of patients actively followed each month and the corresponding retention rates up to 12 months of age. Of these 552 patients, 139 (25.2%) were evaluated by a team of pediatric gastroenterologists for having two or more signs or symptoms suggestive of CMA according to the telephone survey performed by our nurse coordinator. After being evaluated by the specialist, 49/139 infants (35.3%) met the criteria for suspected CMA and underwent an exclusion diet followed by open OFC. CMA was confirmed in 25 infants, representing 4.5% of the cohort and 51% of the cases that underwent the complete process of CMA diagnostic workup. Two additional patients exhibited a marked clinical improvement following the exclusion diet. However, their parents refused to proceed with the OFC due to concerns about the potential severity of a reaction. Given the nature of their symptoms, their severity, and the strong response to dietary exclusion, the evaluating pediatric gastroenterologist ultimately classified these cases as CMA. Overall, the estimated prevalence was 4.9% (95% confidence interval 3.1–7.0%), considering 27/552 subjects (Figure 1).

Table 1 shows the sociodemographic and perinatal characteristics of the infants and the atopy history of their parents in the whole cohort, comparing patients categorized as CMA and otherwise healthy infants. No difference in perinatal history and mode of delivery was observed. The frequency of formula ingestion during the newborn period was low and similar in both groups. The frequency of pets at home was higher (*p* = 0.03), and atopy history in the mother (*p* = 0.04) and anaphylaxis frequency in the father (*p* = 0.049) were lower in the CMA group compared to healthy infants. There was a trend of a higher frequency of asthma in the mothers of healthy infants compared with those with CMA (*p* = 0.05).

The median age of symptomatology onset was 3 months (IQ range 1 to 9.5 months). In 21/27 cases of CMA (74%), diagnosis was achieved before 6 months of age, in 4 cases between 6 and 8 months, in 1 case at 10 months, and in 1 case at 11 months.

Table 2 shows the clinical presentation of the CMA cases. None of the cases had symptoms suggestive of IgE-mediated allergy, and, consistent with that, specific IgE measures were not indicated. Only three infants presented two symptoms; all the others presented at least three symptoms suggestive of CMA when diagnosed. The most frequent symptoms were vomiting, dermatitis, pathologic colic, perianal erythema, and mucous stools. Only five patients had bloody stools.

The 22 patients who were initially suspected of CMA but ruled out by OFC were monitored monthly by phone calls up to 12 months of age, and none of them showed any additional symptoms of CMA during that period. Appendix A presents the final diagnoses of the 90 cases in which a pediatric gastroenterologist ruled out CMA.

At the age of 1 year, all 27 patients with CMA underwent a new open OFC, and 11 (40%) infants tolerated milk.

## 4. Discussion

### 4.1. Prevalence and Comparison with Other Populations

The prevalence of CMA confirmed by OFC in our cohort was 4.5% and increased to 4.9%, including the two cases in which a good response to the exclusion diet was observed, but the parents refused an OFC. This rate exceeds that reported in the Europrevall study but is comparable to other similarly designed studies from Europe and North America, which report rates of 2–7% [7]. Overall, Europrevall reported an incidence of 0.5% in infants, with higher rates in the Netherlands and the UK, where a 1% incidence was reported [10]. This prevalence is also higher than the 1.2% described by Mehaudy et al. in a retrospective Argentinian cohort [20]. In this prospective cohort, we carefully looked for the whole spectrum of gastrointestinal, dermatological, and respiratory symptoms suggestive of CMA, probably increasing the sensitivity of detection of milder cases that could be underrepresented in other studies. This finding suggests that CMA is a significant health concern not only in high-income countries, but also in low- and middle-income regions such as Chile. The consistency of our prevalence data with those from high-income nations underscores the global nature of CMA. It highlights the need for widespread awareness and standardized diagnostic protocols across diverse populations. The prevalence found in this cohort of patients, sampled from an urban low-income population, contrasts with the 9.2% prevalence previously described by our group with a similar methodology, but in a high-income population from the same city [16]. This difference may be partially explained by the demographic and environmental differences between the cohorts. Compared with the current low-income cohort, that group had a higher frequency of cesarean delivery (32%), a group of infants born between 34 and 36 weeks of gestation (8%), and higher parental ages (a mean of 33 years for both mothers and fathers). Additionally, they reported lower pet exposure at home (31%) and a higher frequency of self-reported maternal food allergies (9%). These factors, particularly mode of delivery, early microbial exposure, and maternal atopic history, may influence early immune development and allergen sensitization, contributing to the higher prevalence observed in the high-income group.

### 4.2. Clinical Presentation and Immunological Phenotype

Interestingly, all confirmed cases in our study presented with clinical features compatible with non-IgE-mediated CMA. Although this may initially seem unexpected, similar findings have been reported in other prospective studies in early infancy, particularly when most cases are mild to moderate in severity [21,22,23]. The predominance of non-IgE-mediated presentation in our cohort is likely related to the natural history of CMA in the first months of life and aligns with the updated international MAP (iMAP) guidelines, which recognize that most CMA cases encountered in primary care during infancy are non-IgE-mediated and can be reliably identified through dietary exclusion and reintroduction [24]. In our study, symptoms were systematically assessed through structured monthly interviews, including gastrointestinal, dermatological, and respiratory manifestations. Specific IgE testing and skin prick tests were reserved for patients with clinical features suggestive of immediate hypersensitivity; since no infants met this criterion, these tests were not indicated. The absence of acute-onset symptoms, hallmarks of IgE-mediated allergy, may be explained by the natural history of CMA in early infancy and the genetic or environmental characteristics of our cohort, which included a high proportion of exclusively breastfed infants and delayed exposure to intact CMP. Mild immediate-type reactions may have gone unrecognized by caregivers or resolved spontaneously, thus not triggering further evaluation or referral. These considerations highlight the diagnostic complexity of mixed or atypical presentations in epidemiological studies based on clinical criteria and reinforce the value of symptom-based diagnostic strategies for non-IgE-mediated CMA, particularly in low-resource settings.

### 4.3. Diagnostic Challenges and Methodological Approach

Non-IgE-mediated CMA presents significant diagnostic challenges due to the nonspecific nature of its symptoms and the absence of confirmatory biomarkers such as serum-specific IgE blood tests or skin prick tests. Although the double-blind, placebo-controlled oral food challenge (DBPCOFC) is considered to be the gold standard for confirming diagnosis [1,9], its complexity limits feasibility in routine care and epidemiological studies. In our study, a 4-week exclusion diet followed by an open OFC was employed as the diagnostic strategy, as the aim was to estimate prevalence under real-world clinical conditions. This approach is supported by current consensus guidelines [1,9,25] and is consistent with practical diagnostic algorithms for low-resource settings. Additionally, to minimize interobserver variability, we implemented a standardized symptom checklist for both the nurse-led screening and specialist evaluation.

The high adherence to the protocol (47/49 families agreed to the OFC, 95.9%) reflects strong physician–family communication and a shared understanding of the diagnostic process. Our methodology, combining structured follow-up, standardized symptom assessment, and pragmatic OFC-based confirmation, proved robust for identifying non-IgE-mediated CMA in a real-world setting. Given that most confirmed cases in our cohort were mild to moderate in severity, a high index of suspicion among primary care professionals is essential for timely detection. In this context, the use of a standardized symptom checklist administered by trained nurses demonstrated a good performance as a first-line screening tool and represents a feasible strategy for broader implementation. These findings reinforce the importance of structured clinical protocols and highlight the need for a multidisciplinary approach to CMA management. Nutritionists and dietitians play a key role, particularly in supporting breastfeeding mothers on exclusion diets, by ensuring the complete elimination of CMP from all dietary sources, including processed foods, and preventing nutritional deficiencies, with special attention paid to protein and calcium intake. They also provide critical education on label reading and food preparation practices to avoid cross-contamination. Pharmacists contribute by identifying hidden milk proteins in medications and advising on safe alternatives. The coordination of these healthcare professionals is essential to ensure accurate diagnosis, nutritional adequacy, and effective caregiver education, especially in settings with limited resources.

In breastfed infants, maternal CMP elimination diets were an integral part of our diagnostic protocol, as proteins ingested by the mother and transferred through breast milk can elicit symptoms in sensitized infants. This approach, endorsed by current guidelines [1,9,25], allowed for a non-invasive but clinically informative step in the diagnostic process. Our findings reinforce the relevance of considering maternal diet as both a potential source of allergen exposure and a therapeutic target during the diagnostic phase.

### 4.4. Clinical Implications and Symptom Diversity

The various symptoms observed reflect the diverse clinical manifestations of non-IgE-mediated CMA. This underscores clinicians’ need to consider CMA in the differential diagnosis of a wide range of gastrointestinal and dermatological symptoms in infants [1,8,9].

The early onset of symptoms, with an average age of 3 months and with 74% of infants diagnosed before six months of age, highlights the importance of the early identification and management of CMA to prevent prolonged exposure and associated morbidity. This age range overlaps with the peak of infantile functional gastrointestinal disorders, with which CMA symptoms may co-exist or be confounded. On the other hand, nearly half of the suspected CMA cases in our cohort were ruled out, emphasizing the need for confirmation by OFC in these cases to avoid overdiagnosis.

### 4.5. Oral Tolerance and Outcome at 12 Months

The concept of oral tolerance is fundamental in the natural history of CMA, particularly in non-IgE-mediated forms, where resolution often occurs within the first years of life. In our cohort, re-evaluation with an OFC at 12 months of age provided a practical measure of acquired tolerance under real-world conditions. The 40% tolerance rate to milk at one year of age in our study is lower than the 81% reported by Calle et al. in Peruvian infants with predominantly non-IgE-mediated CMA [22], but is similar to data described in Europe and the United States [26].

### 4.6. Strengths and Limitations

This study has several strengths, including its prospective cohort design, the use of standardized diagnostic criteria with OFC confirmation, and high adherence to follow-up procedures. To our knowledge, it is the first population-based birth cohort in Latin America to estimate CMA prevalence using this methodology. However, some limitations should be acknowledged. The study was conducted at a single public hospital serving a low-income urban population, which may limit its generalizability. Additionally, diagnosis was based on clinical presentation without routine measurements of serum-specific IgE, which may have led to an underrepresentation of IgE-mediated or mixed-type CMA. Finally, the follow-up period was limited to the first year of life, and longer-term outcomes could not be assessed.

## 5. Conclusions

Our study provides valuable insights into the prevalence and clinical characteristics of CMA in a cohort of Chilean infants from an urban, low-income area. A key strength of this study is its robust design. To our knowledge, this is the first population-based cohort study in Latin America with systematic follow-up and standardized diagnostic confirmation of FAs in infancy. Notably, all confirmed cases exhibited non-IgE-mediated presentations, reinforcing the predominance of this form in our population. The prevalence of CMA was comparable to that reported in high-income countries, underscoring the global burden of this condition. These findings emphasize the public health significance of CMA, the need for a high index of clinical suspicion, and the importance of implementing standardized diagnostic protocols that include OFC confirmation to ensure accurate diagnosis and management across diverse populations.

Future research should focus on identifying specific risk factors for non-IgE-mediated CMA and developing more accessible and reliable diagnostic tools. Additionally, exploring the genetic and environmental determinants of non-IgE-mediated CMA in diverse populations could provide valuable insights into the pathogenesis and management of this condition.

## Figures and Tables

**Figure 1 nutrients-17-01859-f001:**
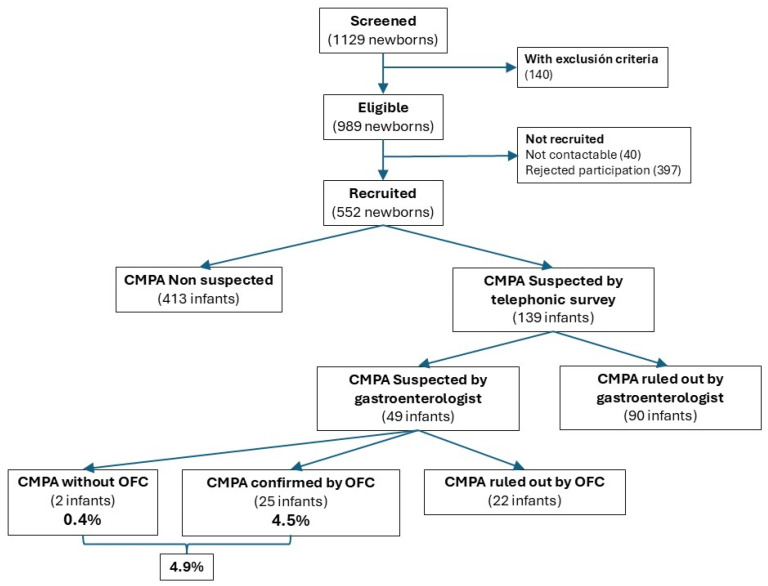
Flowchart of study enrollment, evaluation, and diagnostic confirmation of cow’s milk protein allergy (CMA).

**Table 1 nutrients-17-01859-t001:** Sociodemographic and clinical characteristics of the cohort and comparison between healthy infants and cow’s milk allergy (CMA) cases.

		Overall Cohort N = 552	Healthy Infants N = 525	CMA Patients N = 27	*p*-Value
Newborn history	Gestational age (week) *	39.0 ± 1.1	39.0 ± 1.3	39.0 ± 1.1	0.823
	Male gender, n (%)	265 (48.0)	251 (47.8)	14 (51.9)	0.697 †
	Weight (g) *	3448 ± 474	3442 ± 478	3537 ± 397	0.218
	Height (cm) *	49.7 ± 2.0	49.7 ± 2.0	49.6 ± 2.1	0.705
	APGAR (1 min), median (IQ range)	9 (9–9)	9 (9–9)	9 (9–9)	0.702
	APGAR (5 min), median (IQ range)	9 (9–9)	9 (9–9)	9 (9–9)	0.939
	Cesarean delivery, n (%)	128 (23.2)	119 (22.7)	9 (33.3)	0.240 †
	Exclusively breastfed on discharge, n (%)	508 (92.0)	488 (93.0)	23 (85.2)	0.131 †
Mother history	Age (yrs) *	26.1 ± 6.5	26.1 ± 6.6	26.3 ± 6.6	0.858
	Self-reported atopy, any trait, n (%)	152 (27.5)	140 (26.7)	12 (44.4)	0.049 †
	Rhinitis, n (%)	34 (6.2)	34 (6.5)	0 (0.0)	0.399 †
	Dermatitis, n (%)	25 (4.5)	23 (4.4)	2 (7.4)	0.349 †
	Asthma, n (%)	8 (1.4)	6 (1.1)	2 (7.4)	0.054 †
	Food allergy, n (%)	18 (3.3)	17 (3.2)	1 (3.7)	0.600 †
	Anaphylaxia, n (%)	3 (0.5)	3 (0.6)	0 (0.0)	1.0 †
	Tabaquism, n (%)	182 (33.0)	171 (32.6)	11 (40.7)	0.404 †
	Maximum educational level achieved				
	None, n (%)	13 (2.4)	13 (2.5)	0 (0.0)	0.293 †
	Primary, n (%)	150 (27.2)	138 (26.3)	12 (44.4)	
	Secondary, n (%)	346 (62.7)	333 (63.4)	13 (48.1)	
	Graduate University, n (%)	43 (7.8)	41 (7.8)	2 (7.4)	
	Without remunerated job, n (%)	293 (53.1)	279 (53.1)	14 (51.9)	1.0 †
Father history	Age (yrs) *	28.6 ± 7.3	28.6 ± 7.3	28.5 ± 7.7	0.858
	Self-reported atopy, any trait, n (%)	91 (16.5)	87 (16.6)	4 (14.8)	0.670 †
	Rhinitis, n (%)	31 (5.6)	31 (5.9)	0 (0.0)	0.390 †
	Dermatitis, n (%)	19 (3.4)	19 (3.6)	0 (0.0)	0.616 †
	Asthma, n (%)	3 (0.5)	2 (0.4)	1 (3.7)	0.140 †
	Food allergy, n (%)	15 (2.7)	14 (2.7)	1 (3.7)	0.511 †
	Anaphylaxia, n (%)	1 (0.2)	0 (0.0)	1 (3.7)	0.049 †
	Tabaquism, n (%)	266 (48.2)	250 (47.6)	16 (59.3)	0.323 †
	Maximum educational level achieved				0.363 †
	None, n (%)	19 (3.4)	19 (3.6)	0 (0.0)	
	Primary, n (%)	173 (31.4)	161 (30.8)	12 (44.4)	
	Secondary, n (%)	292 (52.9)	281 (53.5)	11 (40.7)	
	Graduate University, n (%)	67 (12.2)	63 (12.0)	4 (14.8)	
	Without remunerated job, n (%)	41 (7.4)	39 (7.4)	2 (7.4)	1.0 †
Other antecedents	Pet at home, n (%)	352 (63.8)	337 (64.2)	15 (55.6)	0.41 †
	Rurality, n (%)	2 (0.4)	1 (0.2)	1 (3.7)	0.096 †
	Family income per capita (USD) *, **	146.5 ± 78.5	147.5 ± 79.1	129.5 ± 64.4	0.260

(*): Mean ± SD; (**): An answer was obtained in 77.9% of cases. Continuous variables were analyzed using the Mann–Whitney U test. (†) Dichotomic variables were analyzed using Chi-square test, or Fisher’s exact test, when appropriate.

**Table 2 nutrients-17-01859-t002:** Clinical characteristics of infants with cow’s milk allergy.

Symptoms	N (%) N = 27
Vomits and/or frequent regurgitations	23 (85.2)
Atopic dermatitis	12 (44.4)
Pathological colic	11 (40.7)
Perianal erythema	9 (33.3)
Mucus stools	9 (33.3)
Constipation	8 (29.6)
Bloody stools	5 (18.5)
Diarrhea	4 (14.8)
Failure to thrive	2 (7.4)

## Data Availability

The data presented in this study are available on request from the corresponding author due to ethical reasons.

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
