# Peer review of "Prevalence of Cow’s Milk Allergy in Infants from an Urban, Low-Income Population in Chile: A Cohort Study"

_nutrients, 2025, doi:10.3390/nu17111859_

Round 1
Reviewer 1 Report
Comments and Suggestions for Authors
Dear authors, I have reviewed the manuscript entitled “Prevalence of cow's milk protein allergy in infants from an urban, low-income Latin American population: a cohort study.” Overall, the topic is relevant. However, several methodological improvements and stylistic corrections are needed before the manuscript can be considered for publication.
Specific comments:
- In my opinion, the Introduction section is too short and it would benefit from a deeper discussion of the state of art. Please improve it.
- I would suggest dividing “Materials and Methods” Section into subparagraphs (e.g. 2.1. Study Design, etc)
- Please add the protocol number of the study to the “Materials and Methods” Section.
- The manuscript lacks a detailed explanation of the exclusion process. Please clarify it.
- A prevalence of 3% was used for sample size calculation, but no rationale was provided. Please add it.
- Line 97: Please remove the letter “e” before “Table 1”.
- P-values should be formatted in italic (e.g., p < 0.05). Please ensure consistency throughout the manuscript.
- Line 89–91: The use of "an alpha of 0.05" is correct, but please confirm that all statistical tests are two-tailed, and clarify if correction for multiple comparisons was performed.
- Please use bold for “Figure 1, Table 1, etc“ in the titles of the latter. Example: “Table 1. Sociodemographic and clinical characteristics of the cohort and comparison between healthy infants and CMPA cases”.
- Table 1: Please clarify what constitutes “atopy” in parents (self-reported or clinically diagnosed?).
- I would suggest including the structured checklist used by nurses to screen for CMPA symptoms as a supplementary file.
- Given that all cases were non-IgE-mediated, this unusual result should be further discussed.
- The discussion should better highlight the potential global implications of the findings, particularly in promoting the use of OFC protocols in low-resource settings.
- The English is understandable but needs careful proofreading.
- Line 152: Please ensure abbreviations are used consistently. Once introduced (e.g., CMPA), the full form should not be repeated unnecessarily.
- The “References” Section does not follow the MDPI’s guidelines. Please modify it.

Author Response
Comment 1: “In my opinion, the Introduction section is too short and it would benefit from a deeper discussion of the state of art. Please improve it.”
Response: Thank you for your thoughtful observation. We agree that the Introduction was too brief. We have revised and expanded this section to provide a more thorough overview of the current literature on food allergy and cow’s milk protein allergy (CMPA), with particular emphasis on the diagnostic challenges in infancy and the lack of standardized prevalence data in developing countries.
Comment 2: “I would suggest dividing ‘Materials and Methods’ Section into subparagraphs (e.g., 2.1. Study Design, etc).”
Response: Thank you for this helpful suggestion. We have reorganized this section into subheadings to improve readability and clarity, following your proposed structure.
Comment 3: “Please add the protocol number of the study to the ‘Materials and Methods’ Section.”
Response: We have added the ethical approval protocol number in the IRB-Ethical statement.
Comment 4: “The manuscript lacks a detailed explanation of the exclusion process. Please clarify it.”
Response: Thank you for noting this omission. We have now added a detailed description of the exclusion criteria, specifying the medical conditions and logistical factors that led to exclusion.
Comment 5: “A prevalence of 3% was used for sample size calculation, but no rationale was provided. Please add it.”
Response: We appreciate your observation. We have now added the rationale for using a 3% estimated prevalence in our sample size calculation, based on previous reports from similar populations and local clinical experience.
Comment 6: “Line 97: Please remove the letter ‘e’ before ‘Table 1’.”
Response: Thank you for catching this typographical error. It has been corrected as suggested.
Comment 7: “P-values should be formatted in italic (e.g., p < 0.05). Please ensure consistency throughout the manuscript.”
Response: We appreciate this formatting suggestion. We have reviewed and corrected the formatting of all p-values throughout the manuscript to ensure consistency and compliance with journal style.
Comment 8: “Line 89–91: The use of ‘an alpha of 0.05’ is correct, but please confirm that all statistical tests are two-tailed, and clarify if correction for multiple comparisons was performed.”
Response: Thank you for pointing this out. We have clarified in the “Statistical Analysis” section that all statistical tests were two-tailed. Since our analyses were exploratory and comparisons were limited, no correction for multiple comparisons was applied.
Comment 9: “Please use bold for ‘Figure 1, Table 1, etc.’ in the titles of the latter.”
Response: Thank you for your formatting recommendation. We have revised all table and figure titles to follow this formatting convention.
Comment 10: “Table 1: Please clarify what constitutes ‘atopy’ in parents (self-reported or clinically diagnosed?).”
Response: Thank you for this important clarification request. We have now specified in the table and in the methods section that parental atopy was based on self-reported history.
Comment 11: “I would suggest including the structured checklist used by nurses to screen for CMPA symptoms as a supplementary file.”
Response: We appreciate this suggestion and have now included the nurse-administered symptom checklist as a supplementary material file, as requested (supplementary Table 1).
Comment 12: “Given that all cases were non-IgE-mediated, this unusual result should be further discussed.”
Response: Thank you for highlighting this point. We have expanded the discussion to explore possible reasons for the predominance of non-IgE-mediated CMPA in our cohort, including environmental, dietary, and methodological factors, and compared our findings with existing international data.
Comment 13: “The discussion should better highlight the potential global implications of the findings, particularly in promoting the use of OFC protocols in low-resource settings.”
Response: We fully agree and have strengthened the discussion to emphasize the global relevance of our findings, including the importance of using standardized OFC protocols in low- and middle-income countries to ensure accurate diagnosis and appropriate clinical management of CMPA.
Comment 14: “The English is understandable but needs careful proofreading.”
Response: Thank you for your feedback. The manuscript has been thoroughly revised for grammar and style by the corresponding author using the Software Grammarly. Additionally, we used ChatGPT (OpenAI) to assist in improving clarity and fluency. Finally, all content was again carefully reviewed by all the authors.
Comment 15: “Line 152: Please ensure abbreviations are used consistently. Once introduced (e.g., CMPA), the full form should not be repeated unnecessarily.”
Response: Thank you for noting this. We have reviewed the manuscript to ensure that all abbreviations are used consistently and that full forms are not repeated unnecessarily after first mention.
Comment 16: “The ‘References’ Section does not follow the MDPI’s guidelines. Please modify it.”
Response: Thank you for your careful review. We have reformatted all references to fully comply with MDPI’s citation style and referencing guidelines (ACS style).

Reviewer 2 Report
Comments and Suggestions for Authors
This interesting study aimed to establish the prevalence of CMPA in a cohort of infants living in an urban, low income area from the Metropolitan Region of Chile and determine its underlying mechanism. The results contribute to the understanding of the global frequency and mechanism of CMPA.
The authors reported previously that the prevalence of CMPA in was different in urban low-income and high-income populations from the same city. Would it be possible to differentiate the studied population according to income level to check if a similar relationship can be found or the whole studied population was of uniformly low income?
Minor remarks:
Please state overtly the eligibility and exclusion criteria.
Please format References according to the requirements of the journal
Supplementary Tables (eTable1 and eTable2) are not available.
Line 91:” or a non-parametric test”, which one?
OFC, please explain the acronym on the first use
Supplementary Tables (eTable1 and eTable2) are not provided
Line 91:” or a non-parametric test”, which one?
OFC, please explain the acronym on the first use
Please state overtly the eligibility and exclusion criteria.
Please format References according to the requirements of the journal
Author Response
General Comment:
“This interesting study aimed to establish the prevalence of CMPA in a cohort of infants living in an urban, low income area from the Metropolitan Region of Chile and determine its underlying mechanism. The results contribute to the understanding of the global frequency and mechanism of CMPA.”
Response:
We sincerely thank the reviewer for the positive feedback and thoughtful comments. We appreciate the opportunity to improve the clarity and quality of our manuscript based on your suggestions.
Comment 1:
“The authors reported previously that the prevalence of CMPA was different in urban low-income and high-income populations from the same city. Would it be possible to differentiate the studied population according to income level to check if a similar relationship can be found or the whole studied population was of uniformly low income?”
Response:
We thank the reviewer for this insightful suggestion. In this study, the entire population was recruited from a single public hospital located in an urban, low-income area, and all participants were covered by the public healthcare system. Therefore, income levels were relatively homogeneous, and we were not able to stratify the cohort based on income. This contrasts with our previous study conducted in a private healthcare setting, which involved a higher-income population. We have clarified this point in the revised manuscript (Methods and Discussion section).
Comment 2 (Minor):
“Please state overtly the eligibility and exclusion criteria.”
Response:
Thank you for pointing this out. We have revised the “Materials and Methods” section to explicitly describe both the inclusion and exclusion criteria. These now include medical, logistical, and demographic considerations that defined eligibility for participation.
Comment 3 (Minor):
“Please format References according to the requirements of the journal.”
Response:
We appreciate your attention to detail. All references have been reformatted to fully comply with the MDPI guidelines for citation and referencing.
Comment 4 (Minor):
“Supplementary Tables (eTable 1 and eTable 2) are not available.”
Response:
Thank you for noticing this. The supplementary tables (now we have 3 supplementary tables) have now been correctly uploaded and linked within the manuscript submission system.
Comment 5 (Minor):
“Line 91: ‘or a non-parametric test’, which one?”
Response:
Thank you for your observation. We have now specified that the Mann–Whitney U test was used for comparing continuous variables that did not follow a normal distribution.
Comment 6 (Minor):
“OFC, please explain the acronym on the first use.”
Response:
We agree and have revised the manuscript to define OFC (oral food challenge) upon first mention in the Introduction section.
Reviewer 3 Report
Comments and Suggestions for Authors
Abbr Title: Prevalence of CMA in infants from Chile (note…the focus is Chile…not all of Latin America)
General Comments:
This is a well-written manuscript; CMA, particularly in pediatric populations, is critical for us to diagnose and provide appropriate interventions within this population, even among breastfed patients. Maternal elimination diets and implications on infant allergens were not discussed. Oral tolerance was not introduced…this is an important component to be considered in food allergy management. (see Longo G, Barbi E, Berti I, et al. J Allergy Clin Immunol. 2008;121(2):343–347; Meyer T, Venter V. et al. World Allergy Organization (WAO) Diagnosis and Rationale for Action against Cow's Milk Allergy (DRACMA) Guideline update – VII – Milk elimination and reintroduction in the diagnostic process of cow's milk allergy. World Allergy Organization Journal. 2023. 16(7):100785. doi: 10.1016/j.waojou.2023.100785)
Abstract:
L17-35: Good synopsis of study. The authors should have qualified the infant population better by indicating allergen issues among the respective parents.
Note: most of the allergenicity literature abbreviates CMPA as CMA
Introduction:
This section should address CMA associated with breastfed infants, including issues with the respective mothers’ dietary patterns and possible interventions to eliminate food allergens, including bovine milk.
Materials and Methods
L61: This section should address CMA associated with breastfed infants, including issues with the respective mothers’ dietary patterns and possible interventions to eliminate food allergens, including bovine milk.
L80: the statistical approach is acceptable
Results:
L128: Good Table 1; the caption should indicate the data are means SD, not in the table per se (just above L130
L144: Surprised that none of the infants appeared to present hematochezia; urticaria (aka hives) … maybe included with contact dermatitis or atopic dermatitis?
Discussion:
L151: Based on the presented study, this section is reasonable. It could be more complete if the authors were to discuss maternal elimination diets and oral tolerance approaches. A short discussion on IgE and non-IgE mediated responses would have enriched the manuscript.
L198: The authors make brief comments on physicians’ responsibilities; what about nutritionists’ responsibilities and even those of dietitians and pharmacists?
Conclusions:
L222: Without data from all of Latin America, what makes Chile unique relative to CMA?
Author Response
General Comment:
“This is a well-written manuscript; CMA, particularly in pediatric populations, is critical for us to diagnose and provide appropriate interventions within this population, even among breastfed patients.”
Response:
We sincerely thank the reviewer for the positive feedback and valuable recommendations to improve the scientific quality and clinical relevance of our manuscript. We have carefully addressed each of the suggestions below.
Comment 1: “Title: Prevalence of CMA in infants from Chile (note…the focus is Chile…not all of Latin America)”
Response:
Thank you for this clarification. We have revised the title to more precisely reflect the geographic scope of the study. The new title is:
“Prevalence of Cow’s Milk Allergy in Infants from an Urban, Low-Income Population in Chile: A Cohort Study.”
Comment 2: “Maternal elimination diets and implications on infant allergens were not discussed. Oral tolerance was not introduced…this is an important component to be considered in food allergy management.”
Response:
We thank the reviewer for this insightful comment. As correctly noted, the diagnostic protocol in our study included the indication of a cow’s milk protein (CMP) elimination diet for breastfeeding mothers in cases where CMPA was suspected in exclusively breastfed infants. This was described in the Methods section; however, we agree that this point can be made more explicit. We have therefore revised the corresponding subsection (2.4 Diagnostic Protocol for CMPA) to more clearly state the rationale and role of maternal exclusion diets in the diagnostic process.
Regarding oral tolerance, while this was not a primary endpoint of our study, all infants with confirmed CMPA were followed longitudinally up to 12 months of age, and tolerance to cow’s milk was reassessed through reintroduction at that time. This approach is aligned with international recommendations and provides initial insight into the natural course of CMPA. We have clarified this process in both the Methods and Discussion sections to highlight its relevance to food allergy management.
Comment 3: Abstract: “The authors should have qualified the infant population better by indicating allergen issues among the respective parents.”
Response:
Thank you for this thoughtful suggestion. We agree that including information on parental atopic history adds relevant context to the infant population. We have therefore revised the Abstract to indicate that sociodemographic and atopic data were collected from parents at enrollment and that selected variables were compared between infants with and without CMPA. This addition provides a clearer characterization of the cohort.
Comment 4: “Note: most of the allergenicity literature abbreviates CMPA as CMA.”
Response:
We thank the reviewer for this helpful observation. Following your suggestion, we have revised the manuscript to consistently use the abbreviation “CMA” throughout the text, in alignment with standard terminology used in the allergenicity literature.
Comment 5-Introduction: “This section should address CMA associated with breastfed infants, including issues with the respective mothers’ dietary patterns and possible interventions to eliminate food allergens, including bovine milk.”
Response:
We thank the reviewer for this important observation. We agree that CMA in exclusively breastfed infants, and the role of maternal dietary elimination, is a relevant aspect of both diagnosis and management. Although this concept was operationalized in our methodology, we acknowledge that it had not been clearly addressed in the Introduction. In response, we have added to the introduction a sentence explicitly addressing the potential transmission of cow’s milk proteins through breast milk and the role of maternal exclusion diets, with reference to current international guidelines.
Comment 6-Methods: “This section should address CMA associated with breastfed infants, including issues with the respective mothers’ dietary patterns and possible interventions to eliminate food allergens, including bovine milk.”
Response:
Thank you for pointing this out. We have now clarified in the “Materials and Methods” section that when CMA was suspected in breastfed infants, a cow’s milk protein-free diet was prescribed to the mother as part of the diagnostic elimination phase.
Comment 7: Statistical approach acceptable
Response:
We thank the reviewer for this positive assessment. In this revised version of the manuscript, we have added a few clarifying details regarding the specific non-parametric test used, as well as confirmation that all statistical tests were two-tailed. These additions aim to enhance the transparency and reproducibility of the analysis.
Comment 8-Table 1 caption: “The caption should indicate the data are means ± SD, not in the table per se.”
Response:
Thank you. We have edited the table caption accordingly to indicate that data are expressed as means ± standard deviation when applicable.
Comment 9- Line 144: “Surprised that none of the infants appeared to present hematochezia; urticaria (aka hives)… maybe included with contact dermatitis or atopic dermatitis?”
Response:
We thank the reviewer for this thoughtful observation. In response, we clarify that some infants in our cohort did present with hematochezia and rectorrhagia, which were initially grouped under the term “rectal bleeding.” Following your suggestion, we have revised the terminology to “bloody stools” to encompass both presentations more accurately and improve clinical clarity. Regarding urticaria, we confirm that no cases with this symptom were observed during the study period.
Comment 10 – Discussion L151: “It could be more complete if the authors were to discuss maternal elimination diets and oral tolerance approaches. A short discussion on IgE and non-IgE mediated responses would have enriched the manuscript.”
Response:
We thank the reviewer for this valuable suggestion, which has helped strengthen our Discussion. We have now included a brief discussion on maternal elimination diets in breastfed infants, as well as the concept of oral tolerance assessed at 12 months through a follow-up OFC. Additionally, we expanded on the distinction between IgE- and non-IgE-mediated CMA, highlighting the clinical characteristics and epidemiological relevance of both. These additions appear in the revised Discussion section.
Comment 11 – Discussion L198: “The authors make brief comments on physicians’ responsibilities; what about nutritionists’ responsibilities and even those of dietitians and pharmacists?”
Response:
We appreciate this point and fully agree that the multidisciplinary approach is essential. We have expanded this paragraph in the Discussion to highlight the roles of nutritionists, dietitians, and pharmacists in the diagnosis, treatment, and long-term monitoring of infants with CMA.
Comment 12 – Conclusion L222: “Without data from all of Latin America, what makes Chile unique relative to CMA?”
Response:
We thank the reviewer for this observation. We would like to clarify that our statement did not intend to imply that Chile is unique in terms of CMA prevalence or pathophysiology. Rather, we were referring specifically to the design of the study. To our knowledge, this is the first published population-based birth cohort in Latin America that systematically followed infants during their first year of life and confirmed CMA diagnosis using standardized international criteria, including oral food challenge. We have revised the sentence in the Conclusions section to improve clarity and prevent misinterpretation.
Reviewer 4 Report
Comments and Suggestions for Authors
This is an interesting prospective study with adequate novelty. However, some points should be addressed.
- In line 23 of Abstract, please add the word "and" before term newborns.
- In Abstract, the number of participants as well as the male-female percentages should be included in its Methods section.
- The Introduction is too short and it should be at least double by adding more information in the field of the article such as: a) A brief desription of food allergies in infance, b) More details concerning the prevelance of CMP allergy in low-, middle- and high- income countries., c) the predominant immune mechanisms (IgE or non-IgE mediated) should be described at a molecular level should be described, more references should be added.
- At the end of the Introduction section, the authors should emphasize the literature gap that their research study aims to cover.
- The Materials and Methods section should be better organized. The 1st and the 2nd paragraphs of the Results section should be removed at the beggining of the Methods section with a subheading "Study population enrollment" or it could be incorporated in the study design sectio.
- The inclusion and exclusion criteria should be better described in the Methods section.
- In lines 169-170, the authors reported that: This difference could be explained by ethnic, environmental, dietary, and microbiota reasons, among other factors.". This general statement should be more specific by adding more information.
- The 3rd paragraph of the Discussion section needs more analysis based on the provided references.
- In the 4th paragraph of the Discussion section, a comparison analysis of the results of the present study with previous study should be performed by adding relevant references.
- The 5th paragraph of the Discussion section needs more analysis by adding relevant references.
- Again, the 6th paragraph of the Discussion section needs more analysis by adding relevant references..
- At the end of the Dicussion sections, the strengths and the limitations of the present study should be reported in one separate paragraph.
Author Response
General Comment:
“This is an interesting prospective study with adequate novelty. However, some points should be addressed.”
Response:
We thank the reviewer for the positive overall assessment and for the constructive feedback provided. We have carefully addressed each of your comments, which have helped us significantly improve the clarity, structure, and depth of our manuscript. Please find below our detailed point-by-point responses.
Comment 1, Line 23 of Abstract: “Please add the word 'and' before term newborns.”
Response:
Thank you for noticing this. We have added the word “and” before “term newborns” in line 23 of the Abstract to improve grammar and clarity.
Comment 2, Abstract: Participants and gender distribution: “The number of participants as well as the male-female percentages should be included in its Methods section.”
Response:
We thank the reviewer for this suggestion. While we agree that reporting the gender distribution is relevant, we believe that such demographic characteristics are more appropriately included in the Results section rather than in the Methods, in accordance with standard scientific reporting practices. Therefore, we have added the male-to-female distribution to the Results section of the abstract to enhance clarity and completeness, while maintaining methodological consistency.
Comment 3, Introduction: “The Introduction is too short and it should be at least double by adding more information in the field of the article such as: a) A brief description of food allergies in infancy, b) More details concerning the prevalence of CMP allergy in low-, middle- and high-income countries, c) the predominant immune mechanisms (IgE or non-IgE mediated) should be described at a molecular level, more references should be added.”
Response:
We thank the reviewer for this thoughtful and constructive comment. In response, we have substantially revised and expanded the Introduction to provide a broader context on food allergies in infancy, including their global prevalence, the heterogeneity of clinical manifestations, and the differential distribution of IgE- versus non-IgE-mediated cow’s milk allergy (CMA) across regions and income levels. These additions aim to better frame the relevance of our study within the global and regional burden of disease.
Regarding the request to include molecular-level immunological mechanisms, we appreciate the suggestion; however, we chose not to expand on this aspect in depth as the primary focus of our manuscript is clinical and epidemiological. Including detailed molecular immunopathogenesis would exceed the scope and objectives of this study. Nevertheless, we have strengthened the distinction between IgE- and non-IgE-mediated mechanisms from a clinical diagnostic perspective, supported by updated references.
Comment 4, Introduction: “At the end of the Introduction section, the authors should emphasize the literature gap that their research study aims to cover.”
Response:
We thank the reviewer for this important observation. We agree that the previous version of the Introduction lacked a clear statement of the specific knowledge gap addressed by our study. In the revised version, we have incorporated a dedicated paragraph highlighting the absence of population-based cohort studies using standardized diagnostic criteria—including OFC—in Latin American settings. We also explain how this gap limits current understanding of the prevalence and clinical expression of CMA in these populations. This addition now more clearly frames the rationale and contribution of our study within the broader context of the existing literature.
Comment 5, Methods: “The 1st and 2nd paragraphs of the Results section should be removed at the beginning of the Methods section with a subheading ‘Study population enrollment’ or it could be incorporated in the study design section. The inclusion and exclusion criteria should be better described in the Methods section.”
Response:
We thank the reviewer for this thoughtful suggestion. In the revised version of the manuscript, we have reorganized and expanded the Methods section to include a clear and detailed description of the inclusion and exclusion criteria, as recommended. Regarding the suggestion to move the initial recruitment flow from the Results to the Methods section, we respectfully opted to retain this content within the Results, as it represents the first data generated by the study and provides a logical starting point for reporting outcomes. We believe this approach maintains clarity while aligning with common practices in cohort study reporting.
Comment 7, Lines 169–170: “the authors reported that: This difference could be explained by ethnic, environmental, dietary, and microbiota reasons, among other factors.". This general statement should be more specific by adding more information.”
Response:
We thank the reviewer for this important observation. In response, we have replaced the general statement with a more detailed explanation based on data from our previous cohort study conducted in a high-income population from the same city. We now describe specific demographic and environmental differences—such as higher cesarean rates, parental age, gestational age, lower pet exposure, and increased maternal food allergy—that may contribute to the observed variation in CMA prevalence. This revised text appears at the end of the first paragraph of the Discussion section.
Comment 8: Discussion: “The 3rd paragraph of the Discussion section needs more analysis based on the provided references. In the 4th paragraph of the Discussion section, a comparison analysis of the results of the present study with previous study should be performed by adding relevant references. The 5th paragraph of the Discussion section needs more analysis by adding relevant references. Again, the 6th paragraph of the Discussion section needs more analysis by adding relevant references.”
Response:
We thank the reviewer for these valuable and detailed suggestions. In response, we have significantly expanded the Discussion section to provide a more in-depth interpretation of our results in the context of published literature. We added new references to support our findings on prevalence, diagnostic strategies, and clinical presentation, and we have clarified how our data align with or differ from those reported in other studies. These additions are intended to strengthen the scientific rigor and contextual relevance of our conclusions. We hope the revised version now meets the reviewer’s expectations.
Comment 12, Discussion: “At the end of the Dicussion sections, the strengths and the limitations of the present study should be reported in one separate paragraph.”
Response:
We thank the reviewer for this important suggestion. In response, we have added a new and dedicated paragraph at the end of the Discussion section summarizing the main strengths and limitations of the study. We believe this addition improves the structure and transparency of the manuscript and provides a balanced view of the findings.
Reviewer 5 Report
Comments and Suggestions for Authors
I congratulate the authors for carrying out this interesting study. However, before it can be considered for publication in Nutrients, I have a few improvement suggestions as follows:
The abstract is well written and illustrates the conducted work. However, I miss some directions for further studies at the end of the Conclusions.
The Introduction has to be considerably improved. Please provide more data on the addressed topics and better justify how your research can be relevant. This section must be expanded.
Divide the Materials and Methods into subsections for better comprehension.
The justification for the sample size and its calculation is not clear. Please elaborate on this. How well does your sample represent the Chilean population?
The Results are well presented, but I suggest improving the Discussion. Divide it into subsections and include more discussions by adding more citations from other relevant investigations carried out in different regions worldwide.
The study’s limitations need to be mentioned and properly discussed.
Future perspectives (Lines 215-219) should be moved to the end of the Conclusions section.
Author Response
General Comment:
“I congratulate the authors for carrying out this interesting study. However, before it can be considered for publication in Nutrients, I have a few improvement suggestions as follows.”
Response:
We sincerely thank the reviewer for the encouraging comments and for the constructive suggestions aimed at improving the quality and clarity of our manuscript. We have addressed each point carefully as detailed below.
Comment 1, Abstract: “The abstract is well written and illustrates the conducted work. However, I miss some directions for further studies at the end of the Conclusions.”
Response:
We thank the reviewer for this thoughtful suggestion. In response, we have added a brief sentence at the end of the Conclusions section of the abstract to indicate future research directions, specifically highlighting the need to investigate long-term outcomes and risk factors associated with non-IgE-mediated CMA. We believe this addition strengthens the final message of the abstract.
Comment 2, Introduction: “The Introduction has to be considerably improved. Please provide more data on the addressed topics and better justify how your research can be relevant. This section must be expanded.”
Response:
We thank the reviewer for this constructive observation. In response, we have substantially revised and expanded the Introduction to provide greater context regarding the prevalence of cow’s milk allergy (CMA) across different income settings and the challenges of clinical diagnosis, particularly in infants. The revised version includes more detailed background information and better highlights the relevance of our study in addressing the lack of population-based data using standardized diagnostic criteria in Latin American countries. We hope these improvements meet the reviewer’s expectations.
Comment 3, Methods: “Divide the Materials and Methods into subsections for better comprehension.”
Response:
We thank the reviewer for this helpful suggestion. In the revised manuscript, the Materials and Methods section has been reorganized into clearly labeled subsections to improve readability and comprehension. We also added further details in several parts of this section to enhance transparency and reproducibility.
Comment 4, Sample size: “The justification for the sample size and its calculation is not clear. Please elaborate on this. How well does your sample represent the Chilean population?”
Response:
We appreciate this important observation. We have clarified that the sample size calculation was based on an expected CMA prevalence of 3%, as estimated from previous reports in comparable populations (including a previous study of our group), with a 95% confidence level, 1.5% precision, and an anticipated loss to follow-up of 17.5%. Regarding representativeness, we acknowledge that our sample was drawn from a single public hospital serving an urban, low-income population in the Santiago Metropolitan Region. While it may not represent the entire Chilean pediatric population, it reflects the demographic and healthcare realities of a large segment of infants in urban areas of the country, considering that 80% of our population is assisted by the Public Health System, similar to other Latin American countries.
Comment 5, Discussion: “The Results are well presented, but I suggest improving the Discussion. Divide it into subsections and include more discussions by adding more citations from other relevant investigations carried out in different regions worldwide.”
Response:
We thank the reviewer for this valuable recommendation. In the revised version of the manuscript, the Discussion section has been reorganized using clear and descriptive subheadings to improve readability and thematic structure. This reorganization aims to guide the reader through the main findings, diagnostic challenges, clinical implications, and broader relevance of the study. We believe this change enhances the clarity and coherence of the Discussion.
Comment 6, “The study’s limitations need to be mentioned and properly discussed.”
Response:
We agree and have added a dedicated paragraph at the end of the Discussion section outlining the main limitations of our study, including its single-center design, limited socioeconomic diversity, and potential underdetection of IgE-mediated cases due to protocol focus.
Comment 7: “Future perspectives (Lines 215–219) should be moved to the end of the Conclusions section.”
Response:
Thank you. We have relocated the content on future perspectives to the final part of the Conclusions section, as recommended.
Round 2
Reviewer 1 Report
Comments and Suggestions for Authors
The authors have made the necessary changes to improve the quality of the work, which is now suitable for acceptance in its current form.
Reviewer 4 Report
Comments and Suggestions for Authors
The authors have significantly improved their manuscript.